# Effect of Dew Point and Alloy Composition on Reactive Wetting of Hot Dip Galvanized Medium Manganese Lightweight Steel

**Tong Yang** [1,2,*] , **Yanlin He** [1,2,*], **Zhang Chen** [1,2], **Weisen Zheng** [1,2], **Hua Wang** [3] and **Lin Li** [1,2]

1   School of Materials Science and Engineering, Shanghai University, Shanghai 200044, China; 18800209842@163.com (Z.C.); wszheng@shu.edu.cn (W.Z.); liling@i.shu.edu.cn (L.L.)
2   State Key Laboratory of Advanced Special Steel, Shanghai University, Shanghai 200044, China
3   Instrumental Analysis & Research Center of Shanghai University, Shanghai 200044, China; hwang225@gmail.com
*   Correspondence: shumailyangt@shu.edu.cn (T.Y.); ylhe@t.shu.edu.cn (Y.H.)

**Abstract:** For its application development, the medium manganese lightweight steel with 3 wt.% and 10 wt.% Mn contents was galvanized in continuous hot dip galvanizing (HDG) simulator and the process parameters on the production line were adopted. Combined with the experimental analysis and thermodynamic calculation, the effect of dew point and alloy composition on the reactive wetting of the steel was investigated. It was shown that MnO existed as a stable oxide for the medium Mn steel with 5 wt.% Al as long as Mn content exceeded 5.1 wt.%. The galvanizability of the steel with 10 wt.% Mn was deteriorated resulting from the formation of a thick and continuous external MnO layer, which had adverse effects on the wettability. MnO particles in the form of unstable phase can be found at the surface of 3Mn steel galvanized at dew point +10 °C. It distributed sparsely and the reactive wettability can be obtained by "bridging connection", which mitigated the damage of external oxidation. Moreover, the lower dew point, the less tendency to form external oxide. Although the decrease of dew point to −30 °C had a certain benefit for coating quality, the galvanizing quality of 10Mn steel could not be improved due to the formation of a thick MnO layer. Therefore, the Mn content played a stronger role than dew point on the reactive wetting of hot dip galvanized medium manganese lightweight steel.

**Keywords:** hot dip galvanizing; zinc coating; medium manganese lightweight steel; reactive wetting; thermodynamic calculation

## 1. Introduction

Improved requirements of safety performance and fuel efficiency have led to a strong interest in advanced high-strength steels (AHSS), especially medium manganese lightweight steels with Al content more than 5 wt.%, which have become a hotspot of research due to their excellent combinations of specific strength and ductility [1]. As is well known, the corrosion resistance of AHSS is determined by its galvanizability during continuous hot dip galvanizing (HDG). However, the challenge of galvanization of medium manganese lightweight steel is due to the alloying elements such as Mn and Al. These elements are easily oxidized during hot dipping galvanizing as they have a high affinity for oxygen. The presence of film-forming or granulated surface oxides, particularly MnO [2–4], leads to a severe deterioration of the liquid zinc wettability, and thus hinders the formation of the Fe−Al−Zn inhibition layer by interfacial reaction and worsens the quality of galvanization.

At present, a series of studies on the hot-dip galvanizing and oxidation behavior of medium manganese steel have been investigated, but there are some contradictory experimental results. Alibeigi

et al. [5] studied the selective oxidation behavior of medium manganese steels with Mn contents of 0.14–5.1 wt.% under different dew points. It was shown that the MnO layer was thickened with the increase of Mn content and the high dew point condition (+5 °C) contributed to improving the wettability by deepening the internal oxidation with the decrease of external oxides. The work of Staudte et al. [6] suggested that if Mn content was more than 3 wt.%, the increase of dew point could hardly improve the wettability. Even a large bare area appeared on the surface when Mn content reached 4.6 wt.% under the high dew point condition. Pourmajidian et al. [7] studied the effect of annealing conditions on galvanizability of 0.1C−6Mn−2Si steel. It was determined that the minimum rate of the surface oxide growth corresponded to the lowest oxygen partial pressure at a dew point of −50 °C. However, the experimental results showed that better wetting of zinc liquid could be obtained at high dew point (+5 °C). Moreover, it was noted that the annealing temperature and atmosphere could lead to the change of oxygen partial pressure ($P(O_2)$) even at the same dew point. The $P(O_2)$ was the key to determining the degree of external oxidation and the type of oxidation products, which also have effect on wettability.

Therefore, a comprehensive study is conducted in the present work, aided by thermodynamic calculation, which could predict the formation of Mn/Al oxides determined by the variation of $P(O_2)$. In order to provide a reference for industrial application, the process parameters on the production line are adopted. The surface oxides formed during annealing in various atmospheric conditions in a galvanizing simulator and the inhibition layer formed at the steel/coating interface during the hot dipping of the steel were analyzed. Combined with the experimental and calculated results, the effect of dew point and alloy composition on the reactive wetting of that steel was described in detail.

## 2. Materials and Methods

The chemical compositions of the experimental medium Mn steels are given in Table 1. The dimensions of all the cold−rolled steel sheets used for the hot dipping galvanization were 220 mm × 120 mm × 1.2 mm. Prior to experiments, these sheets were firstly digested in a 5 vol.% HCl solution for pickling, rinsed in deionized water bath, then polished with 4000 grit SiC paper in order to minimize the influence of surface roughness on the follow-up studies. All the sheets were degreased before loading in the galvanization simulator (IWATANI SURTEC).

**Table 1.** Chemical composition of the experimental steels (wt.%).

| Sample Name | Chemical Composition | | | |
| --- | --- | --- | --- | --- |
| | C | Mn | Al | Fe |
| 3Mn | 0.28 | 3.67 | 5.00 | Bal. |
| 10Mn | 0.24 | 10.46 | 5.00 | Bal. |

The reductive gas used in HDG simulator was composed of 80% $N_2$ and 20% $H_2$. Two dew points of 243 K (−30 °C) and 283 K (+10 °C) were set to explore the effect of oxygen partial pressure on the selective oxidation of Mn and Al. The corresponding values of $P(O_2)$ for each process atmosphere at the same annealing temperature of 1093 K (820 °C) were $3.947 \times 10^{-24}$ and $4.125 \times 10^{-21}$, respectively. The annealing cycle consisted of heating to the annealing temperature of 1093 K (820 °C) at a heating rate of 15 K/s, holding at this temperature for 4 min. Then, sheets were rapidly cooled with a cooling rate of 20 K/s to the dipping temperature of 733 K (460 °C) and dipped in the liquid Zn−0.2 wt.%Al bath for 4 s before a final rapid cooling to room temperature. All galvanized sheets were stored in anti-rust oil after galvanization in order to minimize additional oxidation prior to the analysis of microstructure.

Two types of samples were cut from the parent sheets. Samples for reactive wetting investigation were cut from the location covered with zinc coating, which were wiped with 5 vol.% fuming nitric acid carefully in order to expose the interfacial layer under the zinc overlay. Samples for selective oxidation investigation were cut from the clamped ends, which only participated in the annealing

process without immersing in the liquid zinc. In order to rule out the variation in flatness and coating quality of different location, all the samples were taken from the same area.

The sample surfaces were analyzed in a HITACHI SU−1500 field emission scanning electron microscopy (FE−SEM) (CARL ZEISS MICROSCOPY GMBH, Göttingen, German) for the observation of morphology and size distribution of inhibition layer. The accelerating voltage was 20 keV and the working distance was 6–9 cm. The microstructures of the cross-sectional samples, which were prepared by the focused ion beam (FIB) (FEI, Brno, Czech Republic) technique were investigated by means of JEOL JEM−2100F field emission transmission electron microscope (TEM) (JEOL, Tokyo, Japan). Pt coatings were deposited on the region of interest in order to protect the surface microstructure during Ga−ion milling. The scanning transmission electron microscopy (STEM) (JEOL, Tokyo, Japan) and high-resolution scanning transmission electron microscopy (HR−STEM) (JEOL, Tokyo, Japan) were also used to characterize details of interfacial layers. The phase composition was determined by means of energy dispersive spectroscopy (EDS) using a 1 nm electron beam.

## 3. Results

Figure 1 showed a macroscopic view of the galvanized surface of the steel samples. Each kind of sample was annealed at 1093 K (820 °C) in reductive atmosphere conditions with dew point (DP) of +10 °C or −30 °C, respectively. It can be seen that, for both experimental steels, the coating was easier to be suffered from significant surface defects at high DP +10 °C. Figure 1a showed the densely distributed bare spots, the diameter of which was about 5 mm on the surface of 3Mn +10 °C, indicating the steel with 3%Mn treated in DP +10 °C. However, the surface of 3Mn −30 °C was almost completely coated. For 10Mn steels, the large bare area can be observed on the surface of 10Mn +10 °C and the coating quality of 10Mn −30 °C sample was better. But compared with the 3Mn −30 °C, the surface quality of 10Mn −30 °C was not good enough, due to the appearance of obvious zinc droplets and bare areas.

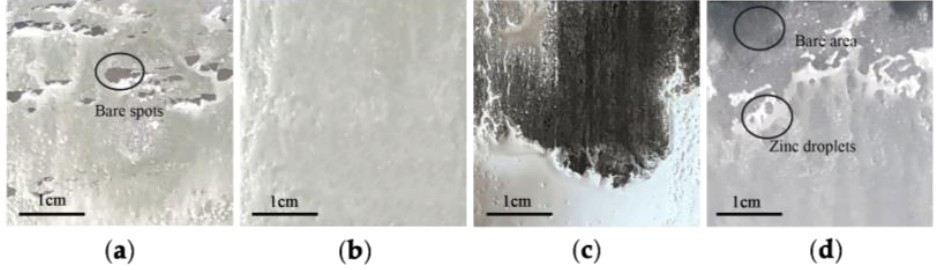

**Figure 1.** Coating quality of the galvanized medium Mn lightweight steel sheets; (**a**) 3Mn +10 °C, (**b**) 3Mn −30 °C, (**c**)10Mn +10 °C, (**d**) 10Mn −30 °C.

Figure 2 showed SEM micrographs of the steel/coating interface at the steel samples stripped with a 5 wt.% HCl solution. The inhibition layer at the steel/coating interface was composed of regular cubic crystals of Fe−Al−Zn phase. In a high DP atmosphere +10 °C, there was an obvious aggregation of the Fe−Al−Zn particles of 3Mn steel, and some of the grain−clusters were bigger than 1 μm in diameter, as shown in Figure 2a. Particles in the inhibition layer of 3Mn in a low dew point −30 °C was tiny with the diameter less than 0.5 μm, as shown in Figure 2b. Compared with the larger area substrate exposed between grains of inhibition layer for the 10Mn sample at the dew point +10 °C, the distribution of Fe−Al−Zn particles in dew point −30 °C was more uniform and compact, as shown in Figure 2c,d. However, at the same dew point condition, the size of particles on the surface of 10Mn samples was bigger than that on the surface of 3Mn sample. It could be concluded that finer and well-distributed Fe−Al−Zn particles were more likely to appear on lower Mn alloys annealed in a lower DP gas atmosphere.

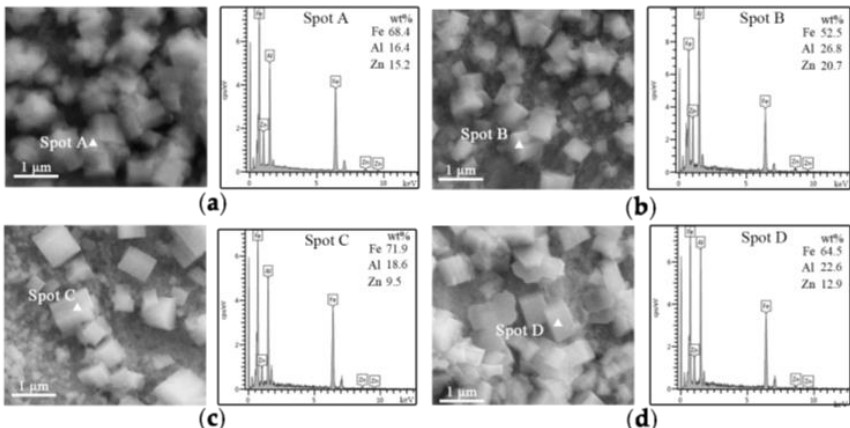

**Figure 2.** SEM micrographs of the surface after removal of the outer zinc layer; (**a**) 3Mn +10 °C, (**b**) 3Mn −30 °C, (**c**) 10Mn +10 °C, (**d**) 10Mn −30 °C.

Figure 3 showed TEM analysis of the coating/substrate interface of the sample 3Mn +10 °C. In order to protect the interface layer from damage, about half of the micron zinc layer was reserved in the FIB sample preparation. It was shown that an obvious MnO particle with diameter of about 200 nm remained at the surface after galvanizing. Around the MnO, mainly Mn−Al oxides were detected. Most of the crystalline $xMnO \cdot Al_2O_3$ was observed internally in the subsurface, which was considered to be harmless to galvanization.

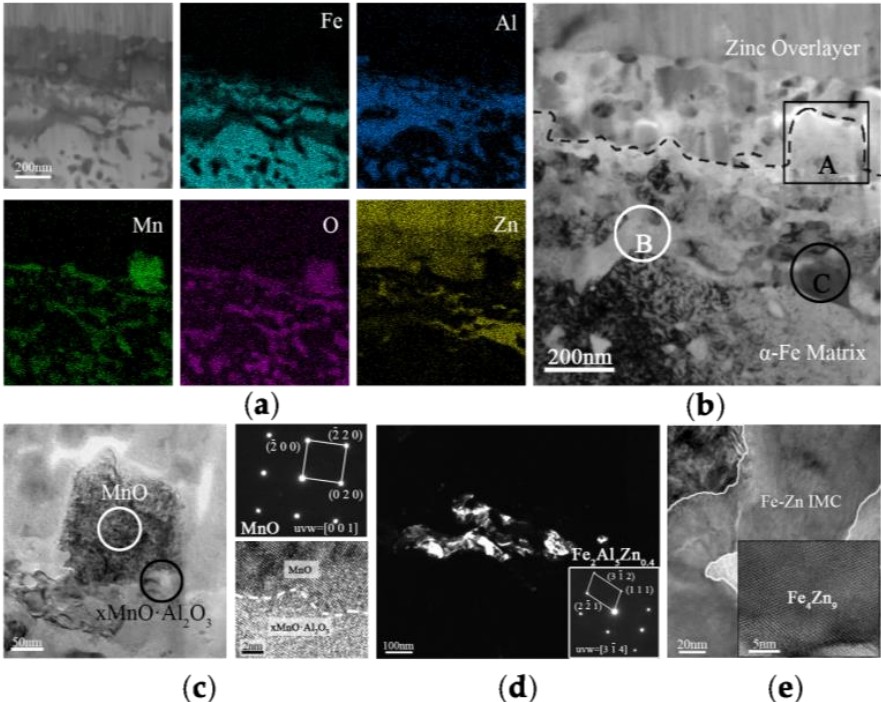

**Figure 3.** (**a**) STEM cross-sectional elemental maps of the coating/substrate interface of the galvanized 3 Mn steel annealed under the 283 K (+10 °C) DP process atmosphere. (**b**) TEM micrograph of the area shown in (**a**). (**c**) Enlargement of the area corresponding to zone A in (**b**). (**d**) Dark filed image and diffraction pattern of the inhibition layer corresponding to zone B in (**b**). (**e**) Enlargement and lattice image of the area corresponding to zone C in (**b**).

In addition, most of $Fe_2Al_5Zn_{0.4}$ inhibition layer could be observed at the grain boundaries and extended into the Fe substrate. The $Fe_4Zn_9$ was adjacent to $Fe_2Al_5Zn_{0.4}$ layer. These results indicated that zinc atoms could diffuse into the iron substrate, and the reduction reaction also happened from

inside, resulting in the laterally growth of inhibition layer under the external oxides. Because of the depletion of Al participating in aluminothermic reduction, part of the Fe−Al−Zn phase transformed into Fe−Zn phase.

Figure 4 showed TEM analysis of the coating/substrate interface of the sample 3Mn −30 °C. The microstructure of the whole interface layer differed from the abovementioned 3Mn sample, which was annealed at a high dew point. The external oxide layer mainly consisted of tiny Mn−Al oxide, which was no bigger than 50 nm in diameter. In the peripheral region of these $xMnO \cdot Al_2O_3$, $Al_2O_3$ was observed instead of MnO, which presented in the similar position on 3Mn +10 °C.

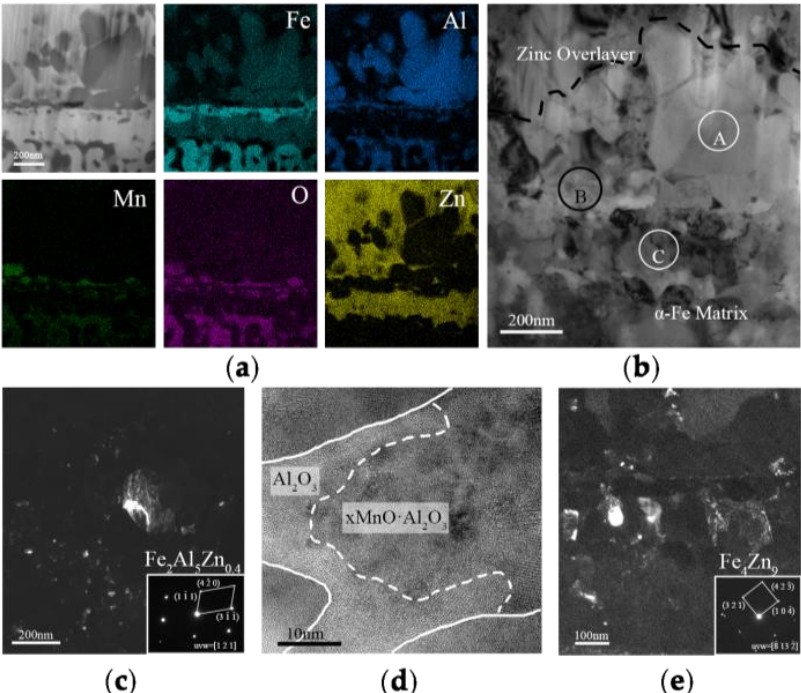

**Figure 4.** (**a**) STEM cross-sectional elemental maps of the coating/substrate interface and subsurface of the galvanized 3Mn steel annealed under the 243 K (−30 °C) DP process atmosphere. (**b**) TEM micrograph of the area shown in (**a**). (**c**) Enlargement of the inhibition layer corresponding to zone A in (**b**). (**d**) Lattice image of the interface between the amorphous $Al_2O_3$ and the crystalline $xMnO \cdot Al_2O_3$ corresponding to zone B in (**b**). (**e**) Enlargement and lattice image of the area corresponding to zone C in (**b**).

Meanwhile, compared with the 3Mn +10 °C sample, the inhibition layer consisting of Fe−Mn−Al presented a more continuously compact distribution on the interface of 3Mn −30 °C sample, as shown in Figure 4a. Most of the $Fe_2Al_5Zn_{0.4}$ particles were 50–200 nm in diameter and gathered at the upper side of external oxide rather than subsurface, and it formed a relatively continuous and dense morphology. There was still an obvious $Fe_4Zn_9$ layer located in the matrix. It is supposedly caused by the diffusion of zinc atoms across grain boundaries.

Figure 5 showed TEM analysis of the coating/substrate interface of the sample 10Mn +10 °C. The formation of continuous 400–600 nm thick layer of MnO can be observed at the coating/substrate interface. The external MnO layer was so integrated that it was hard for liquid zinc to contact the Fe matrix and formed the inhibition layer. However, it was noted that there was an $Al_2O_3$ layer with about 5 nm thickness presented at the interface between zinc coating and MnO. According to the following reaction [8], the $Al_2O_3$ layer may be a reduction reaction product, so the zinc coating could still attach to the $Al_2O_3$ layer.

$$\frac{2}{3}Al + MnO \rightarrow Mn + \frac{1}{3}Al_2O_3, \tag{1}$$

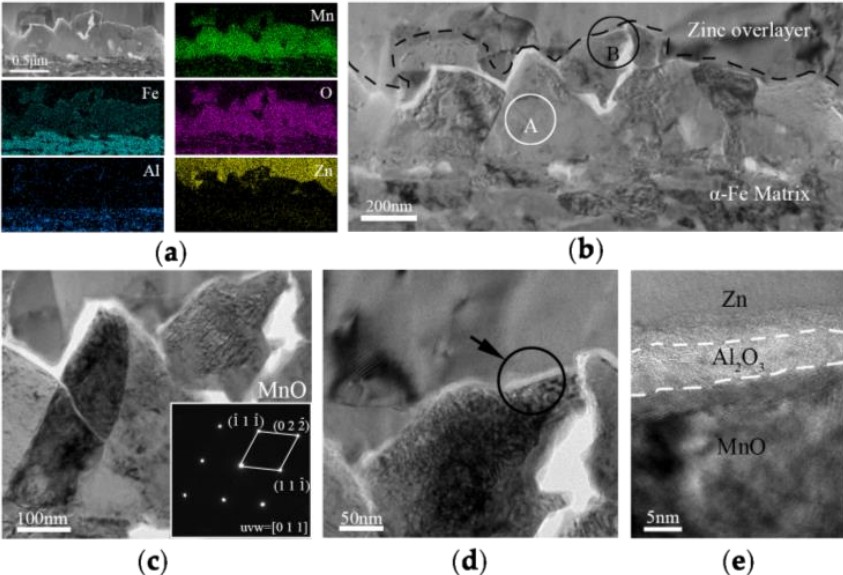

**Figure 5.** (**a**) STEM cross-sectional elemental maps of the coating/substrate interface and subsurface of the galvanized 10Mn steel annealed under the 283 K (+10 °C) DP process atmosphere. (**b**) TEM micrograph of the area shown in (**a**). (**c**) Enlargement and diffraction pattern of the MnO corresponding to zone A in (**b**). (**d**) Enlargement of the area corresponding to zone B in (**b**). (**e**) Enlargement of the area corresponding to circle region in (**d**).

Figure 6 showed TEM analysis of the steel/coating interface of the sample 10Mn −30 °C. The external oxide was mainly composed of a discontinuous but relatively dense MnO layer in thickness of 100 nm. The thin $Al_2O_3$ layer was presented beneath the reduced Fe. Though it was located very close to MnO layer, there was only a little part of them combined and formed $xMnO·Al_2O_3$ composite oxides. In addition, a little bit of $Al_2O_3$ was also located inside zinc coating. It could be the reduction product of aluminothermic reduction. It was noted that the $Fe_2Al_5Zn_{0.4}$ particle 0.5−1 μm in diameter can be observed at the coating/substrate interface.

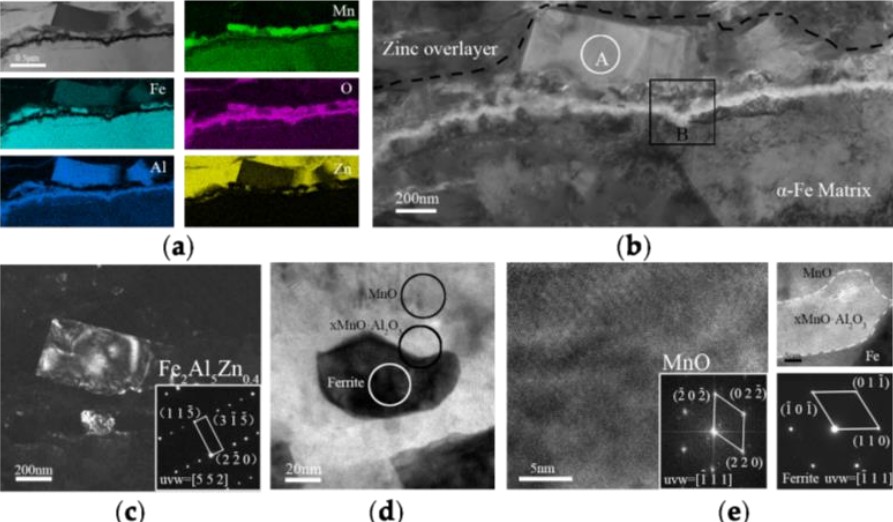

**Figure 6.** (**a**) STEM cross-sectional elemental maps of the coating/substrate interface and subsurface of the galvanized 10Mn steel annealed under the 243 K (−30 °C) DP process atmosphere. (**b**) TEM micrograph of the area shown in (**a**). (**c**) Dark filed image and diffraction pattern of the inhibition layer corresponding to zone A in (**b**). (**d**) Enlargement of the area corresponding to zone B in (**b**). (**e**) Detailed lattice information of MnO, $xMnO·Al_2O_3$, and Fe corresponding to circle regions in (**d**).

## 4. Discussion

### 4.1. Thermodynamic Calculation on the Formation Oxides

Due to the short isothermal holding times and the complex thermal cycle used in heat treatments, the selective oxidation behaviors during continuous annealing in an HDG line was a complex process. Therefore, it was difficult to analyze the mechanism of the selective oxidation process for the hot dip galvanized medium manganese lightweight steel. In the present work, aided by thermo-calc software, the effect of dew point and the Mn content on the thermodynamic stability of oxides were investigated based on the work of Suzuki et al. [9]. In their thermodynamic model of selective oxidation, it was assumed that the local thermodynamic equilibrium was established between the oxide layer and metal ion diffusion layer during annealing process. In addition, the thermodynamic data were provided by the self-built database [10–12] and SSUB6, respectively. The oxide stability determined by $P(O_2)$ corresponding to different Mn content could be calculated, as shown in Figure 7.

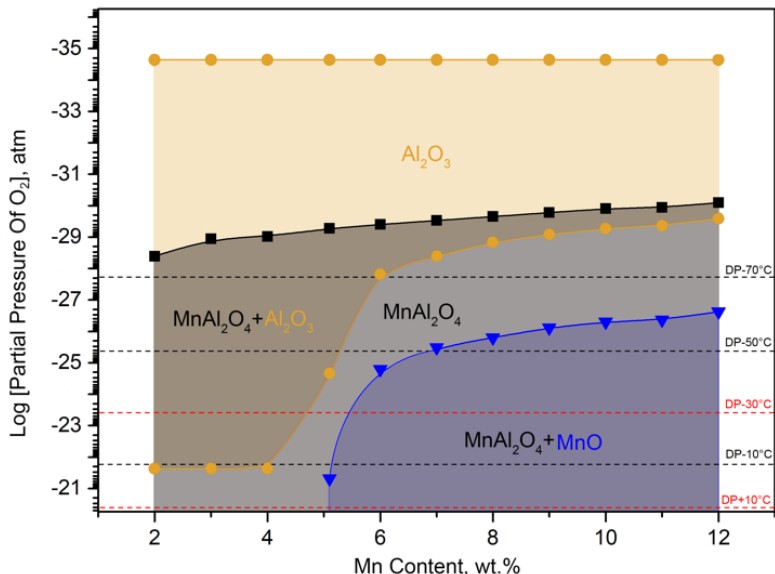

**Figure 7.** The dependence of the oxide stability on the equilibrium oxygen partial pressure and Mn content for the medium manganese lightweight steel.

From Figure 7 and Table 2, it was shown that the stable oxides could be formed on the surface of each kind of experimental steel under the corresponding conditions. It could be seen that $MnAl_2O_4$ was the most stable surface oxides. This kind of composite oxide composed of alumina oxide and manganese oxide was reported by Bellhouse. [13], which appeared in the form of $xMnO \cdot Al_2O_3$. There was a reaction for the composition of the $MnAl_2O_4$ [14]:

$$MnO(s) + Al_2O_3(s) \rightarrow MnAl_2O_4(s). \tag{2}$$

**Table 2.** Stable oxides under different dew points by calculation.

| Sample Name | Dew Point | |
| --- | --- | --- |
| | DP −30 °C | DP +10 °C |
| 3Mn | $MnAl_2O_4$, $Al_2O_3$ | $MnAl_2O_4$ |
| 10Mn | $MnAl_2O_4$, MnO | $MnAl_2O_4$, MnO |

For 10Mn steels, MnO would be the stable external oxide on the surface. From Figure 7, it can be seen that the higher oxygen content, the more MnO content. According to the experimental results as

mentioned above, from Figures 3–6, MnO particles remained much less for 3Mn steels than 10Mn steels. Thus, the overall coating quality of 10Mn steels must be worse than 3Mn steels under corresponding conditions because MnO layer led to a severe deterioration of the liquid zinc wettability. It can especially be seen from Figure 7 that the MnO was able to form a stable external oxide layer on the surface of medium Mn lightweight steel with 5 wt.% Al as long as Mn content exceeded 5.1 wt.%. That is to say, MnO was the stable external oxide for the medium Mn lightweight steel above 5 wt.% Mn content during HDG. It was possible to lead the surface coating defect because the MnO layer was too thick to be reduced in aluminothermic reduction reaction.

In order to explain the effect of alloy composition on the type of selective oxidation products under the same DP condition, the mass fraction of Al/Mn in $MnAl_2O_4$ was considered as:

$$Al/Mn = 2 \times 26.982/54.938 = 0.98. \tag{3}$$

Under the thermodynamic equilibrium condition, the effect of Al/Mn on the type of oxide was calculated. As shown in Figure 8, if the Al/Mn ratio in the alloy composition was more than 0.98, excess Al would lead to the consumption of Mn to generate $MnAl_2O_4$, thus the remaining compound may be $Al_2O_3$. It corresponded to the oxidation behavior of 3Mn steels. On the other hand, once the ratio was less than 0.98, the surplus Mn would oxidize into MnO, which is consistent with 10Mn oxidation behavior.

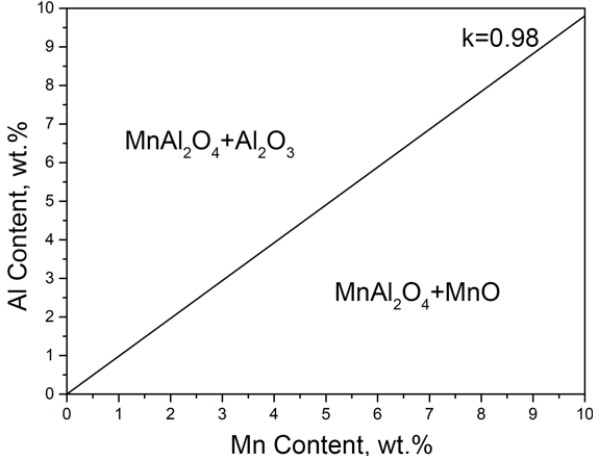

**Figure 8.** Effect of Al/Mn on the type of oxides.

It was noted that the coexistence of $MnAl_2O_4$ and MnO could be observed at the coating/substrate interface of the galvanized 3Mn steel annealed under the +10 °C DP process atmosphere. However, for the 3Mn steel under the −30 °C DP, the coexistence of $MnAl_2O_4$ and $Al_2O_3$ could be found. One possible reason lied in manganese oxides would form as the "unstable oxides" under the non-thermodynamic equilibrium condition due to the short isothermal holding times. Another possible reason was the preferentially generated $Al_2O_3$ formed in deeper layer from the surface because of the higher affinity of Al to oxygen. Moreover, the higher dew point, the less tendency to form $Al_2O_3$ at surface.

### 4.2. Reactive wetting

It was well known that the type, thickness, and morphology of external oxides will influence the reactive wetting behavior. According to the work of Kawano [2] the wetting angle of the zinc solution on MnO layer would be maintained at 114° and not change with the thermite reduction in the zinc bath, so the presence of MnO always had adverse effects on the wettability. The wetting angle of the zinc solution on $Al_2O_3$ was always less than 90°, which seemed to be the static wetting. Moreover, the $Al_2O_3$

tended to form internal oxidation so it had little effect on galvanizability. The $MnAl_2O_4$, as a common oxide on the surface of galvanized CMnAl steels, also has little effect on coating quality [15].

In order to explain the selective oxidation of 3Mn +10 °C and 3Mn −30 °C, the formation of oxide on the surface was also analyzed by TEM, as shown in Figure 9. It was shown that the thickness and morphology of MnO external oxide layer was different comparing the two samples. For the sample annealed at high dew point +10 °C, the thickness of continuous MnO layer was up to 200 nm, and it was hard to be reduced completely during galvanization process in 4 s. Hence, there was still MnO particles remained on the surface of the sample 3Mn +10 °C, as shown in Figure 3. On the contrary, for the 3Mn steel annealed at low dew point −30 °C, the maximum thickness MnO layer was only half that of the former and distributed discontinuously. Obviously, it is easily reduced completely during galvanization. It was known that the thickness and morphology of the oxide film was also the key to galvanizing quality, since the different reactive wetting behavior was observed with the same kind of MnO oxides. It could be further concluded that the high DP atmosphere condition results in the formation of thick film-type external oxides, whereas the low DP atmosphere condition could make the MnO layer thinner and form the discontinuous particle-type external MnO.

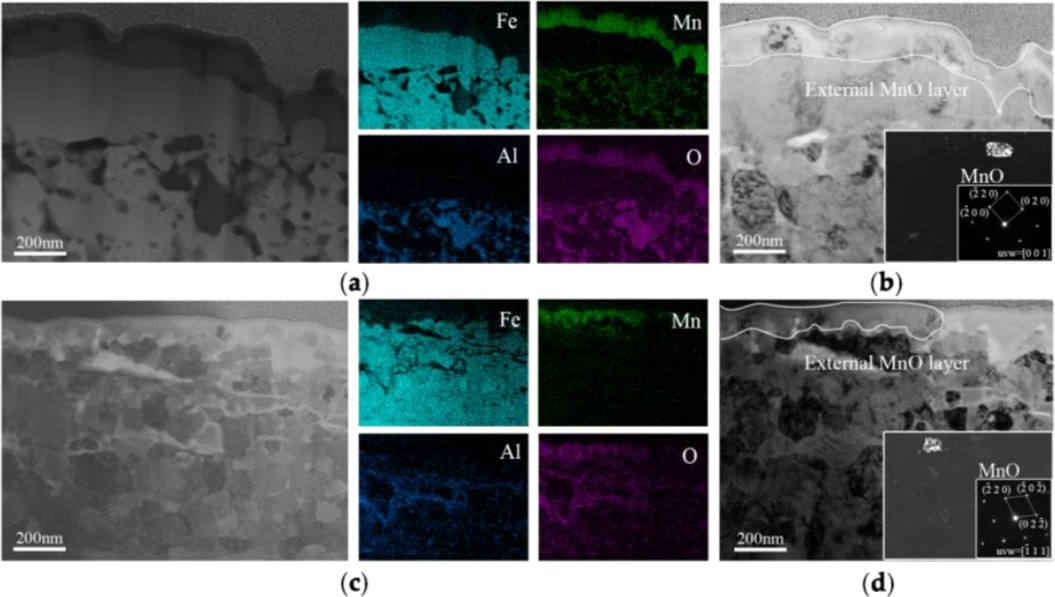

**Figure 9.** Cross-sectional TEM micrographs and EDS elemental distribution maps of the annealed surface of 3Mn +10 °C (**a**,**b**) and 3Mn −30 °C (**c**,**d**) selective oxidation samples.

Khondker et al. [8] showed that the reduction of MnO by the solute Al in the Zn bath was, in principle, thermodynamically possible. According to Kavitha and McDermid recent work [16], the thickness of a MnO layer was significantly reduced for increasing reaction times and the product of the aluminothermic reaction could be detected at the MnO/Zn interface. The abovementioned provided the evidence for the removal of the oxide film by aluminothermic reduction: Compared with the 3Mn oxidation samples shown in Figure 9, the MnO layer of 3Mn galvanizing samples was reduced a lot, especially at the DP −30 °C. For the two 10Mn steel samples, the reduction product $Al_2O_3$, which was observed on the outermost of whole interface layer or inside zinc coating also proved the existence of reduction reaction.

Another way to form the inhibition layer depended on the "bridging connection" through grain boundaries below external oxide layer. Lawrence [17] reported the oxides located on the top of inhibition layer. When the oxide layer was cracked but the iron matrix did not expose enough area to react with the aluminum in the zinc solution, zinc atoms would diffuse across grain boundaries and connected to formed inhibition layer on subsurface. In this study, this kind of formation process of inhibition layer only happened on 3 Mn steels, and it was the main way to form the inhibition layer of

3Mn +10 °C sample. Therefore, it would be likely to generate a lot of brittleness Fe−Zn intermetallic compounds, which may largely deteriorate the mechanical property of medium Mn lightweight steel.

In summary, the reactive wetting mechanism and the comprehensive influence of alloy content and dew points for that steel could be explained by means of the schematic shown in Figure 10.

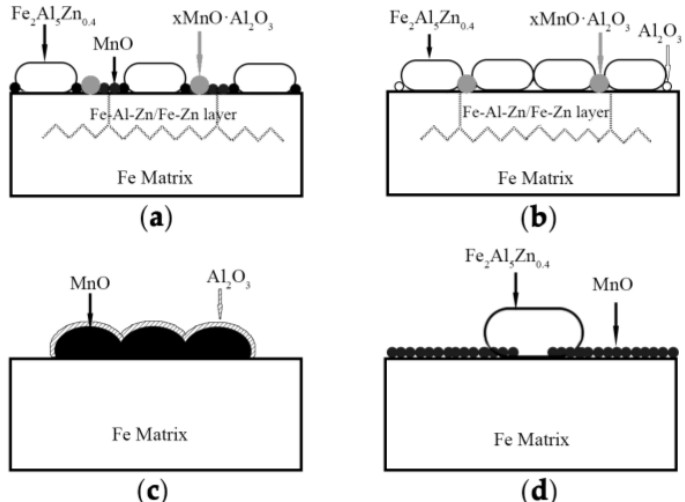

**Figure 10.** Schematic illustration of the reactive wetting influenced by alloy content and dew point during hot-dip galvanizing of medium Mn lightweight steel. (**a**) 3Mn +10 °C, (**b**) 3Mn −30 °C, (**c**) 10Mn +10 °C, (**d**) 10Mn −30 °C.

At the same dew point, however, there exhibited significantly different ways for reactive wetting between 3Mn steels and 10Mn steels by comparing Figure 10a with Figure 10c and Figure 10b with Figure 10d, respectively. For the relatively low Mn content samples, see Figure 10a. Al in zinc bath could partially reduce the MnO, which led to the breaking of the continuity of the external oxide layer. Thus, the small part of Fe was exposed and involved in the reaction to nucleate the $Fe_2Al_5Zn_{0.4}$ grains. As for the remaining MnO particles, though it hampered the reactive wetting, it distributed sparsely so that the zinc atoms diffused into the subsurface through grain boundaries. Such formation of the special structure consisted of Fe layer and Fe−Al−Zn layer resulting from "bridging connection", which improved the good effect of reactive wetting and the galvanizability.

The reactive wetting behavior was quite different for the relatively high Mn content samples, see in Figure 10c. The surface was covered by manganese oxide layer with about 0.5 μm in thickness, and the Fe−Al−Zn grains were dispersedly distributed in the layer. Compared with low Mn content samples, the one possible reason of poor galvanizability for the higher Mn sample lay in the decreasing exposed areas of reduced Fe substrate brought a consequent destruction of inhibition layer. Another possible reason was the absence of "bridging connection" effect, which weakened the degree of reaction wettability. In addition, comparing Figure 10b with Figure 10d, the thicker layer of MnO that formed on the surface of 10 Mn −30 °C resulted in the low nucleation rate and the abnormal grain growth of inhibition layer. For the 3Mn −30 °C, the discontinuous external MnO layer could be completely reduced by aluminothermic reduction and the occurrence of "bridging connection".

For the same manganese content, the ways of reactive wetting at dew point +10 °C and dew point −30 °C were also different. The reactive wetting was improved at low dew point. Because the thin particle-type MnO layer that formed at low dew point can be reduced completely, together with the diffusion of zinc atoms, the uniform growth of the inhibition layer was promoted. For the high Mn sample at low DP (see Figure 10d), though MnO was the stable oxide, the remaining surface oxide after aluminum thermal reduction reaction was far less than that at high dew point. It could be seen that the adverse effect of selective oxidation could be alleviated by lowering the dew point. It was noted that the effect of DP on the galvanizability of 10Mn steel was limited, because the external oxide layer

(stable MnO) was too thick to be broken by reactive wetting. The influence of manganese content was more than that of dew point once the Mn exceeding 5.1wt.% for medium manganese steel, because the thermodynamically stable MnO would form on the surface of the sample with 10wt.%Mn content and led to the deterioration of galvanizing.

## 5. Conclusions

Combined with the galvanization experimental analysis and thermodynamic calculation, the effect of dew point and alloy composition on the reactive wetting of hot dip galvanized medium manganese lightweight steel was investigated. The main conclusions are as follows:

At the same dew point, the ways of reactive wetting comparing 3Mn steels with 10Mn steels were different. According to thermodynamic calculated results, MnO was able to form stable external oxide layer on the surface of medium Mn lightweight steel with 5 wt.% Al as long as Mn content exceeded 5.1 wt.%. The galvanizability of the steel with 10 wt.%Mn deteriorated, resulting from the formation of thick and continuous external MnO layer, which had adverse effects on the wettability. In addition, the formation of oxide could be determined by estimation about the effect of Al/Mn ratio on the type of oxide for the steel. If the Al/Mn ratio in the alloy composition was more than 0.98, $MnAl_2O_4$ or $Al_2O_3$, which had little effect on the wettability would form on the surface of the steel; if the ratio lower than 0.98, the MnO would form, which had worse effect on the respond wettability.

For the same manganese content, the ways of reactive wetting at dew point +10 °C and dew point −30 °C were also different. The MnO particles in the form of unstable phase can be found at the surface of galvanized steel for the 3 Mn steel at +10 °C. However, it distributed sparsely and the reactive wettability can be obtained by "bridging connection" and it mitigated the damage of external oxidation. Moreover, the lower dew point, the less tendency to form external oxide. Although the decrease of dew point to −30 °C had a certain benefit for coating quality, the galvanizing quality of 10Mn could not be improved due to the formation of thick MnO layer. Hence, the Mn content played a stronger role than dew point on the reactive wetting of hot dip galvanized medium manganese lightweight steel.

**Author Contributions:** Conceptualization, Y.H. and T.Y.; methodology, Y.H.; software, T.Y.; validation, T.Y.; formal analysis, H.W.; investigation, T.Y.; resources, Y.H.; data curation, W.Z.; writing—original draft preparation, T.Y.; writing—review and editing, T.Y.; visualization, Z.C.; supervision, L.L.; project administration, Y.H.; funding acquisition, Y.H. All authors have read and agreed to the published version of the manuscript.

**Funding:** This work is financially supported by the National Key R&D Program of China (2017YFB0304402) and the National Natural Science Foundation of China (51971127).

**Conflicts of Interest:** The authors declare no conflict of interest.

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
