# Peer review of "Effect of Dew Point and Alloy Composition on Reactive Wetting of Hot Dip Galvanized Medium Manganese Lightweight Steel"

_coatings, doi:10.3390/coatings10010037_

Round 1
Reviewer 1 Report
The influence of dew point on wetting of medium-manganese steel with liquid zinc was determined in the article on the basis of experimental research and thermodynamic calculations. The issue has a utilitarian character in the processes of continuous galvanizing of sheets by the Sendzimir method, and the use of galvanized sheets with a high manganese content is of great importance mainly in the automotive industry.
The summary sufficiently characterizes the content of the article.
The keywords are correctly selected, but I think it would be beneficial to use the word "hot dip galvanizing", which would greatly facilitate linking the article with a particular technological process.
The introduction well characterizes the current state of knowledge in relation to the literature of the last few years and the legitimacy of the research undertaken. Although I believe that the description of the research methods in the introduction (verses 61-63) is not necessary, because these methods are described in the next chapter.
The methodology for sample creation and preparation as well as the research methodology are described in detail, but minor comments are given below.
Specific comments to the article:
Line 69: Digestion is carried out in the hydrochloric acid solution, not degreasing. In hot dip galvanizing, this process is called picling. However, the process "cleaned with acetone" is better called degreased.
Line 100: the chapter should not start with a drawing only with text with reference to the drawing.
Line 102 "Galvanized surface of the galvanized ............ steel" this sentence is incomprehensible (incorrect).
Fig.1. You can enlarge the drawing by aligning it to the right and left edges of the text
Fig 2. Figure 2 should be placed below the text in which reference is made to this figure. In addition, Figure 2 shows the surface of the coating after removal of the outer layer, and not the interface steel/coating. Fig.2. should be enlarged because it is illegible.
Fig. 3 and 4. Figures 3 and 4 should be placed below the text in which reference is made to this figure. Fig. 3 and 4 should be enlarged because they are illegible.
Line 163-164: Penetration of liquid zinc in grain boundaries is very questionable. The diffusion of zinc atoms is more likely, which can be easier across grain boundaries. This claim requires discussion or reference to literature.
Figure 5. Notes as for Figures 3 and 4.
Figure 6. Notes as for Figures 3 and 4
Table 7 should be below the text in which reference is made to this table.
Figure 8. Notes as for Figure 3 and 4
Figure 9. Fig.9. should be enlarged because it is illegible.
Line 277-278: "the liquid zinc would permeate through grain boundaries". Grain boundaries are defects in the crystallographic lattice. There is no loss of lattice continuity here, only a disturbance of the crystallographic lattice regularity. Zinc atoms can diffuse in the crystallographic lattice, and grain boundaries as defective areas make diffusion much more possible. The penetration of liquid zinc across grain boundaries is rather questionable. Such a statement requires clarification.
Line 295 Note as above
Author Response
Dear reviewer:
Thank you so much for your careful guidance. I am honoured to receive your comments and corrections. The followings are the changes of manuscript according to your opinion:
Adjusting the position of Figure1-9 and Table 2 below the text which referred the Figure/Table. Resizing all the Figures so that they can be more clearer. Revising all the "penetration of liquid zinc" to "diffusion of zinc atoms", I feel very sorry for the gross mistake which should not have happened. And I looked up related literature to ensure this correct statement. Adding keywords; Revising the impropriety of words and the sentences which was not fluent.
Please see the revised manuscript in the attachment.

Reviewer 2 Report
This paper studies the effect of dew point and alloy composition on the reactive wetting of hot dip galvanized medium manganese steel. The authors evaluate two different compositions and perform the process with two different PD. In addition, the study is complemented with thermodynamic calculations to explain the formation of oxides.
The introduction provide useful information for the comprehension of the problem. It presents an evaluation of the previous works of other authors, and clearly states the objective of the research. The methodology used is explained correctly, the results and discussion are clear, and the conclusions are supported by the data previously analyzed.
In my opinion, the paper can be published with the following minor corrections:
Line 76 - use P(O2) instead of p(O2)
Figure 2 - The figure shows an image and a graph of each sample. It would be convenient to present them separately so that the axes of the graph could be seen more clearly

Author Response
Dear reviewer:
Thank you so much for your careful guidance. I am honoured to receive your comments and corrections. The followings are the changes of manuscript according to your opinion:
Resizing Figure 2 so that it can be more clearer. Revising all the "p(O2)" to "P(O2)"
Merry Christmas~
BEST REGARDS
Yang Tong